# Past and Recent Effects of Livestock Activity on the Genetic Diversity and Population Structure of Native Guanaco Populations of Arid Patagonia

**DOI:** 10.3390/ani11051218

**Published:** 2021-04-23

**Authors:** Andrés Mesas, Ricardo Baldi, Benito A. González, Virginia Burgi, Alexandra Chávez, Warren E. Johnson, Juan C. Marín

**Affiliations:** 1Laboratorio de Genómica y Biodiversidad, Departamento de Ciencias Básicas, Universidad del Bio-Bío, Chillán 3780000, Chile; andresmesasp@gmail.com (A.M.); alexandra.chavez.a@gmail.com (A.C.); 2Instituto Patagónico para el Estudio de los Ecosistemas Continentales, Centro Nacional Patagónico, CONICET, Puerto Madryn U9120 ACD, Argentina; baldi.ricardo@gmail.com (R.B.); virburgi@yahoo.com (V.B.); 3Wildlife Conservation Society, Buenos Aires C1426 AKC, Argentina; 4South American Camelids Specialist Group, SSC, IUCN, Santiago 8330015, Chile; bengonza@uchile.cl; 5Laboratorio de Ecología de Vida Silvestre, Facultad de Ciencias Forestales y de la Conservación de la Naturaleza, Universidad de Chile, Santiago 8330015, Chile; 6Center for Species Survival, Smithsonian Conservation Biology Institute, National Zoological Park, 1500 Remount Road, Front Royal, VA 22630, USA; johnsonw11661@gmail.com

**Keywords:** camelids, control region, microsatellites, human effects, genetic structure

## Abstract

**Simple Summary:**

Determining the impacts of human activities on natural populations is important for biodiversity conservation. In this paper, we study the past and more recent effects of urbanization and livestock activity on the genetic diversity and population structure of endemic guanaco populations of the arid Monte and Patagonian Steppe of central Argentina. Our results reveal that urbanization, the installation of fences, and the competition from sheep grazing coincided with the isolation of several guanaco populations, especially in areas with the highest intensity of livestock activity. However, our genetic analyses suggest that a more recent increase in connectivity among groups is occurring. Our results highlight the importance of implementing conservation management plans for natural populations in arid and human-intervened environments.

**Abstract:**

Extensive livestock production and urbanization entail modifications of natural landscapes, including installation of fences, development of agriculture, urbanization of natural areas, and construction of roads and infrastructure that, together, impact native fauna. Here, we evaluate the diversity and genetic structure of endemic guanacos (*Lama guanicoe*) of the Monte and Patagonian Steppe of central Argentina, which have been reduced and displaced by sheep ranching and other impacts of human activities. Analyses of genetic variation of microsatellite loci and d-loop revealed high levels of genetic variation and latitudinal segregation of mitochondrial haplotypes. There were indications of at least two historical populations in the Monte and the Patagonian Steppe based on shared haplotypes and shared demographic history among localities. Currently, guanacos are structured into three groups that were probably reconnected relatively recently, possibly facilitated by a reduction of sheep and livestock in recent decades and a recovery of the guanaco populations. These results provide evidence of the genetic effects of livestock activity and urbanization on wild herbivore populations, which were possibly exacerbated by an arid environment with limited productive areas. The results highlight the importance of enacting conservation management plans to ensure the persistence of ancestral and ecologically functional populations of guanacos.

## 1. Introduction

Understanding the factors and processes linked with the genetic diversity and structure of natural populations is a fundamental goal of evolutionary and conservation biology. Both population diversity and structure are the results of demographic factors, historical biogeography, and life history characteristics [1], and both are crucial to understanding the local adaptations and evolutionary potential of wild populations. However, landscape characteristics that restrict connectivity among populations are usually more closely linked with patterns of genetic variation [2]. For example, landscapes dominated by human intervention, habitat fragmentation, urbanization, artificial barriers, including fences, roads, and land-use patterns, often reduce connectivity and limit gene flow [3,4,5,6]. When sustained effectively for sufficient time, decreased connectivity can then lead to increased population structure and/or inbreeding, ultimately isolating small populations and limiting their adaptive potential [7]. These processes are often exacerbated in arid environments where natural resources are scarce and patchily distributed [3,8].

The arid northern and central regions of Argentine Patagonia are a transition zone between the Monte and Patagonian Steppe phytogeographical provinces [9]. The most distinctive climatic feature of the area is low rainfall (125–250 mm annual average) associated with high interannual variation. The dominant native ungulate of the region is the guanaco (*Lama guanicoe*), which is well-adapted to arid and semiarid environments [10]. The distribution of guanacos has repeatedly been influenced by the effects of global climate change, most recently during the Last Glacial Maximum around 25,000 years ago [11,12,13,14]. It has a wide but discontinuous range, from the Peruvian Andes and the Bolivian Chaco in the north to Tierra del Fuego and Navarino Islands in southern Argentina and Chile, and from sea level to 5000 m in elevation in the Andes Mountains [15]. There are two recognized guanaco subspecies: *L. g. cacsilensis* is distributed mainly from northern Peru to northern Chile and *L. g. guanicoe* from the Bolivian Chaco to the Fuegian archipelago [14]. Despite this broad distribution, their numbers have decreased by 70% [16] since the arrival of Europeans through continuous hunting and the effects of livestock activity [17]. Today, although the most abundant guanaco populations inhabit the Argentinean Patagonia, these have suffered drastic reductions in local distributions and abundance.

In Argentinean Patagonia, livestock and agricultural activity began around 150 years ago along the Chubut river, following the arrival of Welsh settlers to the coastal plains near Puerto Madryn. Through the 1890s, domestic sheep were introduced, and their numbers increased rapidly, reaching a peak of 22 million individuals during the 1950s [18]. Extensive sheep ranching and production of meat and wool led to the establishment of permanent infrastructure, including buildings, fences, and roads that limited the free movement of guanacos and restricted access to more productive foraging areas with greater availability of key plant species [19,20].

Effective management and conservation of livestock and guanaco populations in Patagonia require knowledge of both historical and more-recent population structure and genetic diversity of the guanacos in the region. Previous genetic studies of broad geographic and evolutionary patterns demonstrated that guanaco populations of the Patagonian steppe and Tierra del Fuego [21] were of the *L. g. guanicoe* subspecies, including the Monte desert populations of northern Patagonia in Argentina [14], and found evidence of recent gene flow among the populations of the southern Patagonian steppe with northern and central Patagonia [14].

To more precisely assess the impact of human activities and sheep production in the region on the genetic structure of the guanacos in northern Patagonia, we evaluate patterns of genetic variation of 15 variable microsatellite markers and one mitochondrial gene in guanacos from several locations of the arid Monte and northern Patagonian Steppe (Figure 1). These nuclear and mitochondrial markers have been widely used in other studies, facilitating broader comparisons. Our principal goals are to (1) assess the genetic heritage of guanaco populations, (2) measure genetic patterns relative to other populations, including the occurrence of potential substructure and connectivity level, and (3) document evidence of inbreeding patterns resulting from past isolation and/or low population size.

## 2. Materials and Methods

### 2.1. Sample Collection and DNA Extraction

Guanaco skin from dead animals and fecal samples were collected at seven localities, representing a large part of their distribution in the southern Monte and northern Patagonian Steppe of Argentina (Figure 1 and Appendix A). GPS coordinates were recorded for each sample and preserved in 30 mL of absolute ethanol. All samples were stored at −80 °C in the Laboratorio de Genómica y Biodiversidad, Departamento de Ciencias Básicas, Facultad de Ciencias, Universidad del Bio-Bío, Chillán, Chile. Total genomic DNA was extracted from the skin using the Wizard Genomic DNA Purification Kit (Promega, Madison, WI, USA); using proteinase K digestion and a standard phenol–chloroform protocol [22], DNA from feces was extracted using the QIAamp DNA Stool Mini Kit (Qiagen, Valencia, CA, USA) in a separate non-genetic-oriented laboratory.

### 2.2. Microsatellite Analysis

Fourteen frequently used autosomal dinucleotide microsatellite loci (YWLL08, YWLL29, YWLL36, YWLL38, YWLL40, YWLL43, YWLL46 [23], LCA5, LCA19, LCA22, LCA23 [24], LCA65 [25], LGU49, and LGU68 [26]) were analyzed using well-established standard protocols. Amplification was carried out in a 10-µL reaction volume, containing 50–100 ng of template DNA, 1.5–2.0 mm MgCl2, 0.325 µm of each primer, 0.2 mm dNTP, 1X polymerase chain reaction (PCR) buffer (Qiagen), and 0.4 U Taq polymerase (Qiagen). All PCR amplifications were performed in a PE9700 (Perkin Elmer Applied Biosystems) thermal cycler, with cycling conditions of initial denaturation at 95 °C for 15 min, followed by 40 cycles of 95 °C for 30 s, 52–57 °C for 90 s, 72 °C for 60 s, and a final extension of 72 °C for 30 min. Amplification and genotyping of DNA from fecal samples were repeated two or three times to confirm the repeatability of the results. One primer of each pair was labeled with a fluorescent dye on the 5′-end, and the fragments were analyzed on an ABI-3100 sequencer (Perkin Elmer Applied Biosystems). Data collection, sizing of bands, and analyses were carried out using GeneScan software (Applied Biosystems). Each plate included an allelic ladder that acted as a positive control to facilitate consistent scoring of loci between plates. Each reaction was repeated three times for reproducible heterozygotes and up to seven times for homozygotes and samples exhibiting allelic dropout or false alleles [27]. Consensus genotypes were constructed from the combined results.

We identified samples that came from the same individual by searching for matching microsatellite genotypes using the Excel Microsatellite Toolkit [28] and eliminated samples from the study if they showed more than 85% overlap. We also evaluated the existence of null alleles using the program Micro-Checker v. 2.2.3 [29]. ARLEQUIN 3.5.1.2 software [30] was used to estimate allele frequency, observed heterozygosity (*H_O_*), and expected heterozygosity (*H_E_*). The inbreeding coefficient *F_IS_* was estimated following [31] and using FSTAT 2.9.4 [32].

### 2.3. Mitochondrial DNA

The left domain of the mitochondrial control region (514 bp) was amplified using the camelid and guanaco specific primers LthrArtio (5′-GGTCCTGTAAGCCGAAAAAGGA-3′), H15998V (5′-CCAGCTTCAATTGATTTGACTGCG-3′), and HLoop550G: 5′-ATGGACTGAATAGCACCTTATG-3′ [33]. Amplification was performed in a 50-µL reaction volume with ~30 ng genomic DNA, 1× PCR buffer (8 mM Tris-HCl (pH 8.4), 20 mM KCl (InvitrogenGibco, Life Technologies), 2 mM MgCl2, 25 µM of each dNTP, 0.5 µM of each primer and 0.1 U/µ Taq polymerase (InvitrogenGibco, Life Technologies^®®^, Carlsbad, CA, USA). All PCR amplifications were performed in a PE9700 (Perkin Elmer Applied Biosystems) thermal cycler, with cycling conditions as follows: initial denaturation at 95 °C for 10 min, followed by 30–35 cycles of 94 °C for 45 s, 62 °C for 45 s, 72 °C for 60 s, and a final extension of 72 °C for 5 min. PCR products were purified using the GeneClean Turbo for PCR Kit (Bio101), following the manufacturer’s instructions. Sequence reactions were visualized using an ABI-3100 sequencer (Perkin Elmer Applied Biosystems). Products were sequenced in forward and reverse directions using BigDye chemistry on an ABI Prism 3100 semiautomated DNA analyzer, and consensus sequences were generated and aligned using Geneious v.9.1.5 (Biomatters, Auckland, New Zealand). The final alignment was trimmed to 514 bp, beginning at the 5′ left domain of the d-loop.

The number of segregating sites (S), haplotypes (nh), haplotype diversity (h), nucleotide diversity (π), and the average number of nucleotide differences between pairs of sequences (k) were estimated using ARLEQUIN 3.5.1.2 [30]. A statistical parsimony network was constructed using TCS v1.21 [34], with default settings.

### 2.4. Estimation and Delimitation of Genetic Units and Gene Flow

We used the Bayesian clustering algorithm implemented in STRUCTURE v.2.3.3 [35] to group the samples genotyped with microsatellites into K clusters. We tested values of K between 1 and 10, running STRUCTURE five times for each value of K and using Evanno’s ΔK method to determine the most suitable number of clusters [36]. STRUCTURE was run using the admixture model and correlated allele frequencies, as recommended for populations that are likely to be similar due to migration or shared ancestry [35,37]; 500,000 iterations were used to estimate K after a burn-in period of 50,000 iterations, as is recommended for detecting structure among populations that are likely to be similar due to migration or shared ancestry [35,37].

A Bayesian analysis of population structure that accounts for the geographical distribution of guanaco was performed with the R package Geneland 1.0.7 [38], with 1,000,000 Markov chain Monte Carlo (MCMC) iterations and a 20% burn-in. These parameters were used for five repetitions of K-values (the number of clusters in the data), ranging from 1 to 10. Using the same parameters and the K-values inferred above as a fixed variable, the MCMC algorithm was run 30 times.

To correlate genetic variation with geography, we estimated effective migration surfaces (EEMSs) for the microsatellite datasets [39]. EEMS identifies geographic regions where genetic dissimilarity decays more quickly than expected under a null model of isolation by distance. It relates effective migration rates to expected genetic dissimilarities to identify regions of high or low migration. The program also estimates levels of effective genetic diversity across the landscape on the basis of genetic distances between individuals sampled within a given site (deme). As such, localities with more dissimilar individuals are estimated as regions of higher genetic diversity. The EEMS method produces visualizations that highlight portions of the species’ distribution where population divergence deviates from strict IBD, which, in turn, can be used to identify gene barriers and/or corridors. Divergence is represented as a function of migration rates under a stepping-stone model [40]. Expected genetic dissimilarities are inferred by sample locations and the migration rates within a grid of demes placed across the distribution. Individuals migrate between nearest-neighbor demes with rates varying by location. To show that our results were independent of grid size, we ran each analysis with grids of 400, 500, and 600 demes. For each analysis, we ran three independent chains for 2,000,000 MCMC iterations, sampling every 5000 iterations, with a burn-in of 1,000,000 MCMC iterations from different starting seeds, and compared log posterior plots to ensure convergence. We ran a series of short preliminary runs to choose parameter values that gave acceptance ratios between the desired 20–30%. All nine runs were combined, and graphs were constructed using the rEEMSplots package in R [39]. Prior to the combination, we analyzed all runs separately to ensure individual run convergence and a good fit of the EEMS model.

Recent migration rates among sampling locations were assessed using BAYESASS v.1.3 [41], an approach that does not assume Hardy-Weinberg (H-W) equilibrium within populations. We set delta values for allele frequencies at 0.3 (maximum change between iterations), inbreeding coefficients at 0.30, and immigration rates at 0.18, so that acceptance rates for changes in these parameters fell between 40% and 60% [41]. BAYESASS was run three times with different random seeds to check for results convergence, with 2 million steps as burn-in and 6 million steps of data collection.

We ran a test of isolation by distance (IBD) by comparing the individual pairwise matrix of genetic distances against the corresponding geographical distance matrix using the Mantel test, with 1000 permutations in ARLEQUIN 3.5.2.2 [30] for alternative population grouping and 10,000 permutations in ALLELES IN SPACE [42] to assess significance.

### 2.5. Demographic History

To identify demographic scenarios that are most consistent with current diversity patterns, we used the coalescent-based framework implemented in MSVAR v1.3 [43], using the microsatellite loci of each sampling locality. MSVAR estimates recent effective-population size (N0), ancestral effective-population size (Nt), and time (t) at which the effective-population size changed from Nt to N0. Three independent runs of MSVAR were carried out with a broad range of prior distributions of the model parameters that accounted for the possibility that (1) the populations have remained stable over time (Nt~N0), (2) that there was a bottleneck (Nt > N0), or (3) that there was a population expansion (Nt < N0). Prior distributions were assumed to have log-normal distributions, with a mean and standard deviation for each parameter truncated at zero, following [43]. As the guanaco-specific microsatellite mutation rate is not known, we used a typical range of vertebrate microsatellite mutation rates of 10^−2.5^ and 10^−4.5^ [44,45,46,47]. MSVAR was run for 400 million iterations under each demographic model, discarding the initial 20% of MCMC steps as burn-in. The independent runs were used to estimate the mode of the posterior distributions of each parameter (N0, Nt, and t) and their corresponding 90% highest posterior density interval. A generation length of three years [10,48] was used to rescale the t parameter in years. Convergence of the runs was estimated with Gelman and Rubin’s diagnostic using the CODA library [49] in R [50].

## 3. Results

### 3.1. Genetic Diversity

Among the 102 samples genotyped with microsatellites, we detected an average of 12.4 alleles/microsatellite. The number of alleles per locus ranged from 7 to 25, and 49 alleles were unique to a single locality. Of these, the highest number (20) were present in Loma Blanca (LB). Deviation from H-W equilibrium due to an excess of heterozygotes was found in Loma Blanca (LB), Meseta Somuncura (MS), and Península Valdés (PV), as well as for the LCA22, LCA5, LCA19, and YWLL40 microsatellite markers. Additionally, a significant excess of homozygotes was found in Bajada del Diablo (BD), Las Plumas (LP), and Ameghino (AM) localities and for LCA85, LCA82, YWLL08, and YWLL38 markers. As estimates of genetic diversity excluding these loci were not significantly different from those estimated with all loci, we used all markers in further analyses (Welch-corrected t-test *p*-value > 0.05). We found consistently high levels of genetic diversity (mean expected heterozygosity ranged from 0.7083 to 0.7625) and high values for allelic richness (mean AR ranged from 1.708 to 1.762) (Table 1).

Twelve variable positions (2.3%), 11 transitions, and one transversion from 514 nucleotides and 15 haplotypes were identified in 82 sequences of the 5′ end of the control region of mitochondrial DNA. Haplotype (*h*) and nucleotide diversity (p) estimates are detailed in Table 1, and the distributions of haplotypes within 7 localities are provided in Appendix A. Sequences were deposited in GenBank with accession numbers JX678410.1–JX678455.1 and MW392248–MW392283.

The 82 Control Region sequences were connected through a network, with a maximum of 5 mutational steps (Figure 1). The shallow branches are consistent with a scenario of rapid demographic expansion. Although the network did not completely delineate a genetic partition corresponding with a geographic separation between samples, a single dominant haplotype (Haplotype 1) was widely distributed among all sampling sites (Appendix A), from which related sequences were separated by one or two mutational steps.

### 3.2. Genetic Structure and Gene Flow

On the basis of their geographical origin and their genetic similarity, southern Monte and northern Patagonia guanacos could be further subdivided into three groups. Results of the STRUCTURE analysis indicated that the best clustering solution was K = 3 based on the ΔK method (Appendix A): (1) Loma Blanca (LB); (2) Meseta Somuncura (MS), Telsen (TE), and Península Valdés (PV); (3) Bajada del Diablo (BD), Las Plumas (LP), and Ameghino (AM). Overall, most of the individuals of mixed or incongruent heritage were from Ameghino (Appendix A). There was a close correspondence between the STRUCTURE analysis results (Figure 2A) and the geographical patterns predicted by the posterior probability of Geneland analyses (Figure 2B).

The pairwise FST values ranged from 0.032 between Meseta Somuncura and Loma Blanca to 0.124 between Loma Blanca and Las Plumas. Significant genetic differences were also observed among the three groups identified in the STRUCTURE analysis. Specifically, Group 1 included only Loma Blanca guanacos, Group 2 encompassed individuals from Meseta Somuncura, Telsen, and Península Valdés, and Group 3 consisted of Bajada del Diablo, Las Plumas, and Ameghino individuals (Appendix A). Similarly, the Meseta Somuncura and Península Valdés localities, included in Group 2, showed significant genetic differences from Las Plumas and Ameghino localities, respectively, both of which were part of Group 3 (Appendix A, Appendix A).

Estimates of effective migration surfaces for *L. guanicoe* suggested that there were higher levels of effective migration than would be expected under a model of isolation-by-distance (IBD) between Meseta Somuncura–Telsen–Peninsula-Valdez and Las Plumas–Ameghino and reduced, fine-scale migration between Meseta Somuncura–Telson–Península Valdés and Las Plumas–Ameghino–Meseta Somuncura (Figure 2C). Estimates of effective diversity surfaces for *L. guanicoe* inferred that there was elevated diversity in Ameghino and Bajada del Diablo but reduced genetic diversity in Península Valdés, Loma Blanca, Meseta Somuncura, and Las Plumas (Figure 2D). This differed from estimates of observed heterozygosity that suggested that the highest genetic diversity occurred in Loma Blanca (Table 1). However, these patterns may be artifacts resulting from reduced sample sizes from these smaller geographic areas. Although estimates of genetic diversity for Península Valdés suggested that there was lower than expected diversity in the northern localities (corresponding to slightly lower *H_E_*), there was no specific locality with elevated levels of genetic diversity (Figure 2D). This pattern is consistent with the relatively consistent levels of *H_E_* among sites and suggests that allelic variation was evenly partitioned across sites. The IBD analysis determined that the observed structure among populations was consistent with isolation by distance (correlation coefficient (r) = 0.40, *p* > 0.05 in ARLEQUIN 3.5.2.2, and r = 0.23, *p* < 0.001 in ALLELES IN SPACE; Appendix A), supporting the hypothesis that there have been genetic barriers in the central and northern portions of guanaco distribution (Figure 2D) and a separate and distinct population in the south.

The posterior probabilities (Appendix A) of recent migration among guanaco locations, as estimated using BAYESASS, were generally lower (less than 1%) than expected for Península Valdés (0.24) and Loma Blanca (0.15 and 0.22). Populations that recorded high posterior probabilities were all geographically close to each other and included recent migrants from Península Valdés to the populations of Meseta Somuncura and Telsen and from Las Plumas to the populations of Bajada del Diablo and Ameghino. Recent migrants to and from Loma Blanca were not detected. The Ne estimation of Meseta Somuncura, Telsen, and Peninsula Valdes was moderately high, with a mean of 440 and a 95% confidence interval from 198.96 to infinity. The Ne estimate for the other two groups was low, with a mean of 76.6 in Loma Blanca and a 95% confidence interval between 44.4 to 223.6 individuals and 92.3 individuals in the combined groups of Bajada del Diablo, Las Plumas, and Ameghino, with a 95% confidence interval of 65.2 to 150.6 individuals, confirming that overall population size in the region was relatively low.

### 3.3. Demographic History

The results of the demographic analyses of the microsatellite data using MSVAR under the three demographic models were relatively consistent. The three independent runs for each sampling locality, of (1) no demographic change (Nt~N0), (2) bottleneck (Nt > N0), and (3) population expansion (Nt < N0), resulted in Gelman and Rubin’s statistics convergence analyses of less than 1.2. In all cases, MSVAR detected evidence of major effective-population size declines at each of the localities, which would be consistent with current or recent population sizes based on census estimates (Appendix A and Appendix A). Each of the localities had ancestral effective-population sizes (Nt) in the order of ~8000 individuals, with 95% highest posterior density intervals (HPD) between ~550 and ~77,000 (Appendix A). Across localities, the estimated times of the bottleneck averaged around 1600 years before the present (YBP; HPD ~80–2000 YBP). Following this event, the effective-population size of guanacos was estimated to be, on average, less than 180 individuals (HPD ~14–~1700; Appendix A and Appendix A). The relatively consistent estimates of ancestral effective-population size and dates of a population bottleneck, combined with the results of migration–drift and isolation by distance analyses, would be consistent with the hypothesis that current (modern-day) guanaco populations in the region descended from at least two large ancestral populations that only recently started diverging, probably through genetic drift.

## 4. Discussion

Our results provide an insight into how human activities, in particular extensive sheep ranching and the development of human settlements, could have influenced the population structure of endemic wild species such as the guanaco. Here, we found that the guanacos in the region are genetically diverse, with low levels of inbreeding. However, there is also evidence of a recent genetic structure that is associated with human activities and likely reflects direct and indirect impacts on guanaco populations and their habitat.

STRUCTURE and Geneland analyses of the microsatellite data identified three populations (K = 3). The most northwestern cluster, Loma Blanca, located at the northern limit of the arid Patagonian Steppe, had the highest estimates of genetic variation, evidence of a larger and more diverse population [14]. The second population, located between the Patagonian steppe (Meseta Somuncura and Península Valdés) and Monte (Telsen) ecosystems, showed evidence of asymmetric gene flow among localities (source–sink dynamics) and relatively high estimated effective-population sizes (Ne), suggesting that this was the remnant of a historically much larger population. The third and southernmost cluster, which is also distributed between the Patagonian steppe (bajada del Diablo and Ameghino) and Monte (Las Plumas) ecosystems, had the lowest levels of observed heterozygosity and the highest estimates of inbreeding.

The individuals from all of the localities appear to share a common evolutionary origin, based on their shared ancestral mitochondrial haplotype (Haplotype 1; Figure 1) and evidence of a rapid demographic expansion, suggesting that current clusters descended from one or more larger populations in the past [14] and that a bottleneck event occurred sometime before European colonization of the area. The demographic analysis with MSVAR suggests that extant guanaco genetic variation likely reflects a strong demographic bottleneck that occurred around 1600 YBP (HPD ~80–2000 YBP). This period coincides with two to three immense volcanic eruptions in the Arctic [51], Central America [52], and the western Pacific [53] that led to a decline in global temperatures for a century. Although the only reported evidence of this phenomenon has been from the northern hemisphere, an environmental change of this magnitude likely also impacted Patagonian communities as they recovered from the effects of the Last Glacial Maximum. Although several local or global environmental factors possibly affected the guanaco population during that timeframe [54], the results demonstrate the need for additional biological and geological research to clarify the origin of the bottleneck and its impact on fauna and flora and regional climate patterns. However, it is remarkable that not only do all extant populations seem to have passed through this bottleneck at approximately the same time but that they all had very similar ancestral effective-population sizes (i.e., ~8000—HPD ~550–77,000). It is, therefore, likely that large guanaco populations that were connected to each other occupied a wide area of the Monte and Patagonian Steppe prior to a relatively recent bottleneck that dramatically reduced the effective-population size to ~180 (HPD ~14–1700). The main consequence of this event would have been the fragmentation and increased isolation of previously well-connected guanaco populations and the relatively small effective populations of more-isolated populations that survive today.

This scenario would explain the high effective-population size (Ne) estimates for each of the three genetic groups of (i) LB: 76.6 (95% CI: 44.4–223.6), (ii) MS-TE-PV: 440 (95% CI: 198.9–Inf), and (iii) BD-LP-AM: 92.2 (95% CI: 65.2–150.6), in spite of their past bottleneck events. The Ne values of these Argentine populations are all larger than the estimated guanaco population of the Torres del Paine National Park in Southern Chile (although with slightly different approaches) of 34.3–43.1 [55]. Our results indicate that the population of northern arid Patagonia studied here most likely did not decrease as low as it did in Torres del Paine, where the number of breeding adults was estimated to have been reduced to as few as 70 during the 1970s [55].

Here, we show through nuclear and mitochondrial analyses that the guanaco populations of the southern Monte and northern Patagonian Steppe of Argentina have high levels of genetic diversity and allele-richness and a low level of inbreeding (Table 1). These observed levels are similar to most previously studied guanaco populations, including relict, small populations, such as the threatened population in the Bolivian Chaco (Mesas et al., unpublished results) [14] and the introduced population of the Falkland/Malvinas Islands, which suffered a strong founder effect and severe bottleneck events [21]. The proximal mechanisms that have helped maintain relatively healthy levels of genetic diversity in guanaco populations likely include their behavior and social dynamics, where juvenile males and females are expelled from their family groups, greatly reducing the probability of reproduction among close relatives and subsequent inbreeding [56].

Man-made barriers and human-related activities have likely been the primary cause of the fragmentation and genetic structure currently observed in the guanaco populations in northern Patagonia. Similar processes have been documented in other species [57,58]. Here, our data suggest that the Chubut river alone did not lead to reduced genetic connectivity between the guanacos from Las Plumas and Ameghino. However, the arrival of Europeans would have brought rapid changes in land-use patterns [59]. Indiscriminate hunting and the establishment of forts, farms, and ranches with fences increased notably, which likely would have considerably reduced population sizes and levels of connectivity among the guanacos. Particularly in the Patagonian steppe region, the introduction of sheep led to a drastic shift in the availability of preferred habitats and forage.

Sheep ranching first started during the 1890s in Eastern Patagonia and spread rapidly along the coast and central plateaus [60]. The establishment of a Welsh colony in 1865 along the Chubut river and the subsequent introduction of sheep in Península Valdés triggered the development of the main cities of the area, including Puerto Madryn, Trelew, and Rawson. Despite the availability of relatively few water sources, there was a rapid increase in the numbers and densities of sheep that led to a reduction in the forage available for guanacos. As a result, guanacos were displaced to less productive areas, such as Loma Blanca and Bajada del Diablo [61,62], well removed from the open grassland habitats that they had commonly used in the past [63].

The Las Plumas and Ameghino localities seem to be separated by the more ubiquitous and continuous presence of human settlements along the Chubut river. The river played a key role in the establishment of agriculture and the colonization of inland regions, providing water for irrigation of the fertile land and facilitating the arrivals of settlers and the foundation of Rawson city. The basin of the river Chubut and the railroad between Puerto Madryn and Trelew city became an important agrocommercial route, facilitating the development of the surrounding areas. This urbanization and increased human activity appear to have formed a significant barrier to the dispersal of guanacos, which would otherwise have been able to cross the river readily [64].

Physical barriers have often been linked with reduced connectivity between populations and/or disruption of historical movement/migration of animals at the local level [57,65,66]. Private fenced properties could explain the genetic differentiation between the individuals from Las Plumas and Telsen, which, despite their geographic proximity, showed very little evidence of genetic contact. STRUCTURE and Geneland analyses assigned these populations to totally different genetic groups. In addition, the presence of individuals in Meseta Somuncura and Telsen localities, with a proportion of ancestry from the Península Valdés locality, and individuals in Bajada del Diablo and Ameghino localities, with a proportion of ancestry from Las Plumas locality, is consistent with the movement of individuals within groups during the last few generations. Loma Blanca is the most genetically distinct relative to the other localities. However, there is no clear factor that would have reduced gene flow with the other two groups, suggesting that perhaps their genetic differentiation is linked with the effects of isolation by distance.

Our genetic analyses suggest that the populations of Argentina’s northern Patagonia guanaco populations are not under threat. Our results suggest that in northern Argentinian Patagonia, there is currently gene flow among localities and increasing densities of guanaco and that there has been at least a partial recovery of local populations, increased connectivity, and increased interbreeding. There are several examples of where local interbreeding between two clusters is likely, as the animals in those areas have composite genotypes at intermediate allele frequencies relative to those observed in more northern or southern guanaco clusters. Gene flow among localities, although asymmetric, combined with recent increases in population size, suggest that the guanaco populations in northern Patagonia are slowly recovering from historically low population sizes. For example, here, we documented directional migrants from Península Valdés in the localities of Telsen and Meseta Somuncura, and from Las Plumas in Bajada del Diablo and Ameghino (Appendix A). Physiological and behavioral adaptations of guanaco to arid conditions (see details in [10]) could explain the current population increases despite increased aridity and erosion in the region.

However, prolonged drought, overgrazing by livestock, declining productivity, and demand for wool have caused ranches to reduce activity or close [59,67], which has increased habitat availability for guanacos [68]. Concurrently, increased habitat protection on private ranches and the reduction of sheep numbers and poaching [63,69], which facilitates the movement and increases the survival of migrants, has had positive impacts [70,71]. The absence of heterozygosity excess in Bajada del Diablo–Las Plumas–Ameghino is consistent with a rapid increase in population sizes and the appearance of novel genetic variation.

Although our findings are consistent with the premise that livestock, agricultural activities, and associated urbanization have contributed to the current structure and fragmentation of guanaco populations, it appears that reductions in livestock numbers and related activity in central and southern Patagonia in the last 20 years have benefited guanaco populations by increasing connectivity and facilitating gene flow among previously isolated localities. Further conservation initiatives that reinforce these trends are continuing. For example, during the last decade, groups of sheep-wool producers in Península Valdés and goat herders in Neuquén (northwest of Argentinean Patagonia) are promoting the nonlethal management of native predators, reductions in livestock densities, and the coexistence of native herbivores through certification programs (e.g., [72]). However, poaching and illegal commerce of guanaco meat continue to be widespread, and expanded law enforcement is needed [15]. Overall, our results suggest that the protection of core areas that are linked biogeographically would help sustain large guanaco populations and would provide a promising model for the long-term persistence of guanacos across the managed landscapes of Patagonia.

## 5. Conclusions

Our results suggest that although guanacos may have been non randomly distributed into recognizable populations prior to the arrival of Europeans, the impacts of extensive sheep ranching and the development of human settlements, has continued to influence guanaco population structure. Protection of core areas that are linked biogeographically would help sustain large guanaco populations and would provide a promising model for the long-term persistence of the species across the managed landscapes of Patagonia. Although urbanization and fencing has likely structured natural populations, increased livestock activity has also contributed to decreased connectivity among guanaco populations. However, the implementation of broad regional management and conservation plans of wild populations, especially across the Monte and northern Patagonian Steppe, which are under threat of increasing urbanization and aridity, would help ensure long-term sustainability of functional ecosystems and human livelihoods in the region.

## Figures and Tables

**Figure 1 animals-11-01218-f001:**
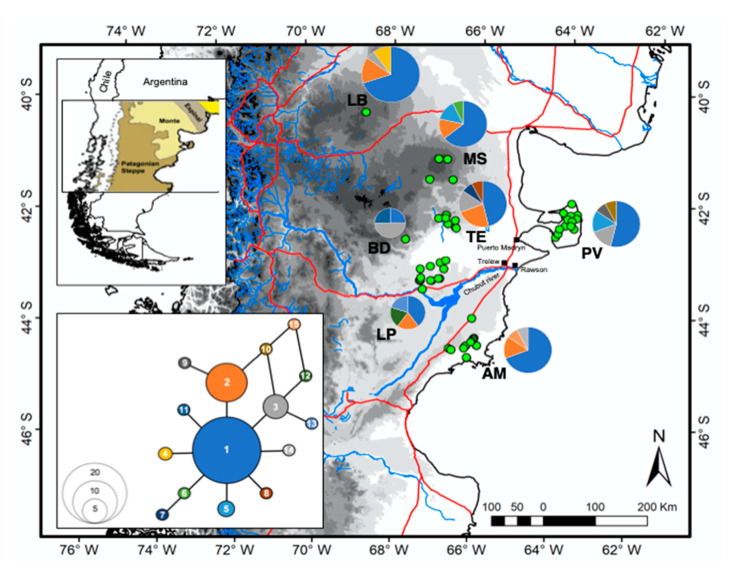
Map of northern Patagonia, with the locations where the individual samples of *Lama guanicoe* used in the analyses were collected represented by green circles. Roads are represented by orange lines and rivers and water bodies by blue lines. Each of the seven locations indicates the relative proportion of each of the 15 haplotypes. Each unique haplotype is represented by a unique color, and the relative proportion of each is depicted in pie charts for each locality. The relative frequency of each haplotype and their phylogenetic relationships are depicted in the insert. The localities’ names are LB = Loma Blanca; MS = Meseta Somuncura; TE = Telsen; PV = Península Valdés; BD = Bajada del Diablo; LP = Las Plumas; AM = Ameghino.

**Figure 2 animals-11-01218-f002:**
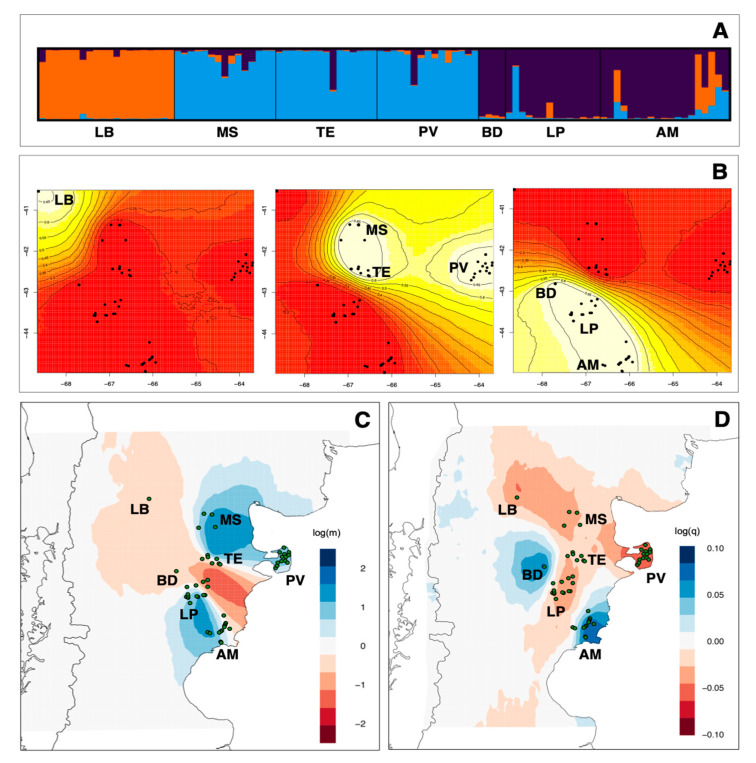
(**A**) Results of cluster analyses of guanacos depicted as a plot of the posterior probability of assignment of 102 guanacos (vertical lines) in three genetic clusters (K = 3) based on Bayesian analysis of microsatellite variation. The individuals are grouped by locality, and localities are indicated along the horizontal axis. The localities’ names are LB = Loma Blanca; MS = Meseta Somuncura; TE = Telsen; PV = Península Valdés; BD = Bajada del Diablo; LP = Las Plumas; AM = Ameghino. (**B**) Results of the posterior probability of Geneland analyses based on patterns of microsatellite variation, with K fixed at 3. Regions with the greatest probability of inclusion are indicated in white, whereas diminishing probabilities of inclusion become from white, yellow to red. Black dots represent sampling sites, and the dashed ovals show the locality groups. Estimates of effective migration (**C**) and diversity (**D**) surfaces inferred by EEMS for the north Patagonia population of *Lama guanicoe*. In (**C**), blue colors represent estimated effective migration greater than would be expected under isolation-by-distance, and darker red colors represent increasingly higher migration. In (**D**), blue colors represent regions of elevated genetic diversity, and increasingly dark red colors represent regions of decreased genetic diversity. Circles represent the four sampling localities in this study.

**Table 1 animals-11-01218-t001:** Summary of the *Lama guanicoe* samples used in the analyses, including localities (ordered approximately from north to south), type of sample (B indicates blood; F, fecal; M, muscle; S, skin; L, liver), and number of samples used from each locality for each genetic marker. Demarcation of regions was assisted by the empirical results of spatial analyses.

Localities	mtDNA					Microsatellites						
	N	*n*	*np*	*h* ± SD	*p* ± SD	N	*A* ± SD	PA	AR ± SD	*H_0_* ± SD	*H_E_* ± SD	*F_IS_* ± SD
Group 1: Loma Blanca	20	4	1	0.5 ± 0.12	0.002 ± 0.001	20	7.533 ± 3.18	20	1.762 ± 0.10	0.905 ± 0.09	0.762 ± 0.1	−0.208 ± 0.17
Group 2						45	8.933 ± 4.86		7.242 ± 3.27			−0.028 ± 0.26
Meseta Somuncura	14	4	1	0.582 ± 0.14	0.002 ± 0.001	15	6.867 ± 3.66	3	1.726 ± 0.22	0.778 ± 0.26	0.726 ± 0.22	−0.082 ± 0.23
Telsen	13	o	2	0.756 ± 0.1	0.004 ± 0.001	15	7.067 ± 2.96	3	1.708 ± 0.13	0.756 ± 0.19	0.764 ± 0.12	0.017 ± 0.21
Península Valdés	13	5	2	0.705 ± 0.12	0.004 ± 0.001	15	6.267 ± 2.96	4	1.764 ± 0.12	0.783 ± 0.25	0.708 ± 0.13	−0.094 ± 0.31
Group 3						37	8.866 ± 3.4		7.512 ± 2.51			0.152 ± 0.22
Bajada del Diablo	4	3	1	0.833 ± 0.22	0.004 ± 0.001	4	3.667 ± 1.11	2	1.722 ± 0.15	0.617 ± 0.29	0.722 ± 0.15	0.128 ± 0.41
Las Plumas	5	4	2	0.900 ± 0.16	0.006 ± 0.001	14	5.933 ± 2.28	4	1.720 ± 0.14	0.673 ± 0.27	0.720 ± 0.14	0.099 ± 0.30
Ameghino	13	4	2	0.526 ± 0.15	0.003 ± 0.001	19	7.333 ± 2.44	13	1.762 ± 0.09	0.635 ± 0.19	0.762 ± 0.09	0.165 ± 0.25

N, number of samples; *n*, number of haplotypes; *np*, number of private haplotypes; *h*, haplotype diversity; π, nucleotide diversity; *A*, number of alleles per locus; PA, private alleles; AR, mean allelic richness; *H_E_*, expected heterozygosity; *H_0_*, observed heterozygosity and their respective standard deviations (SD).

## Data Availability

Data available on request.

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
