# Peer review of "Past and Recent Effects of Livestock Activity on the Genetic Diversity and Population Structure of Native Guanaco Populations of Arid Patagonia"

_animals, 2021, doi:10.3390/ani11051218_

Round 1
Reviewer 1 Report
The authors studied the effect of urbanization and sheep farming on genetic diversity and structure of endemic guanaco and got interesting results. The authors claimed that several populations of guanaco did become more isolated because of human and livestock activities but were recently connected since livestock farming had reduced. This paper provided valued information for conservation managements of wild animals. This paper is well written and organized. I have some major comments:
1) Although three separated population of guanaco can be observed but they could have isolated by other means before the arise of human and livestock activities especially because the heterozygosity of some markers is not low. Moreover, although the population size of guanaco decreased recently, these three separated populations might have existed for quite a long time especially there was a bottleneck before European colonization. Please provide more evidence or explanations.
2) The authors found that the bottleneck occurred around 1,600 YBP but stated that “a relatively recent bottleneck that dramatically reduced the effective population size to ∼180 (HPD ∼14–1,700)”, I found no evidence presented for the “relatively recent bottleneck” in the manuscript, please clarify this.
3) In the Simple Summary, the authors stated: “However, our genetic analyses suggest that a recent decline of livestock activity has improved connectivity among groups of animals.” But this sentence provided less information than the abstract and is even harder to understand. Basically, Simple Summary is for general readers without using jargons, not just simpler than Abstract.
Author Response
Reviewers' comments:
Reviewer #1:
Comments and Suggestions for Authors:
The authors studied the effect of urbanization and sheep farming on genetic diversity and structure of endemic guanaco and got interesting results. The authors claimed that several populations of guanaco did become more isolated because of human and livestock activities but were recently connected since livestock farming had reduced. This paper provided valued information for conservation managements of wild animals. This paper is well written and organized. I have some major comments:
1) Although three separated population of guanaco can be observed but they could have isolated by other means before the arise of human and livestock activities especially because the heterozygosity of some markers is not low. Moreover, although the population size of guanaco decreased recently, these three separated populations might have existed for quite a long time especially there was a bottleneck before European colonization. Please provide more evidence or explanations.
These observations from the reviewer were very useful and we fully agree. We have made some significant changes that we feel help guide the reader through the “narrative”, especially in the Abstract and Discussion. Additionally, we have made an explicit attempt to not over-interpret the data.
2) The authors found that the bottleneck occurred around 1,600 YBP but stated that “a relatively recent bottleneck that dramatically reduced the effective population size to ∼180 (HPD ∼14–1,700)”, I found no evidence presented for the “relatively recent bottleneck” in the manuscript, please clarify this.
We added a sentence that provides a possible explanation to this bottleneck. We have reviewed the entire text and have modified several other sentences to improve clarity.
3) In the Simple Summary, the authors stated: “However, our genetic analyses suggest that a recent decline of livestock activity has improved connectivity among groups of animals.” But this sentence provided less information than the abstract and is even harder to understand. Basically, Simple Summary is for general readers without using jargons, not just simpler than Abstract.
We have edited the Simple Summary and Abstract to make both more informative and to ensure that the “simple summary” is more accessible for the general reader.
Reviewer 2 Report
This study of Mesas et al. explores the genetic diversity of guanacos in Argentina, using mitochondrial control region sequencing and microsatellite genotyping. A total number of 102/82 guanacos have been successfully genotyped/sequenced, from seven locations. The authors’ main interest is the effect of human activities, especially livestock farming and urbanization, on the genetic variation and population structure of central Argentinian guanacos. Although modern sequencing technologies allow sequencing huge parts of the genome, this study is based on the analyses of only 541 bp D-Loop sequences and 14 microsatellite loci. Nevertheless, the employed methods are sufficient for the purpose of the study and more feasible than analyzing larger genomic regions. The conducted analyses, based on the microsatellite and control region data, are straightforward and sophisticated. The results of this study show high genetic diversity and low levels of inbreeding for guanacos in Argentina, suggesting that the guanaco population is recovering. The population has undergone a bottleneck prior to the arrival of European settlers and colonization further decreased the genetic variation. The results are clearly presented and discussed.
Some specific points:
- In line 164 it is stated that the alignment was trimmed to 328 Why? In results 514 bp are mentioned, whereas in Supplementary Table 2 it says 300 bp.
- Table 1: Several columns need to be widened as the numbers/text in them is split over two lines which is confusing and misleading. Also, the table legend mentions ”type of sample” but that is not shown in the actual table.
- Figure 2: The figure legend mentions purple and orange colours in 2C/2D but there is no purple/orange.
- Lines 518-528: shorten sentences.
Minor spelling issues:
- line 50: more closely (no dash)
- line 121: well-established
- line 411: genetic flow
- line 421: larger populations
- line 505: suggest that there has been
Author Response
Reviewers' comments:
Comments and Suggestions for Authors:
This study of Mesas et al. explores the genetic diversity of guanacos in Argentina, using mitochondrial control region sequencing and microsatellite genotyping. A total number of 102/82 guanacos have been successfully genotyped/sequenced, from seven locations. The authors’ main interest is the effect of human activities, especially livestock farming and urbanization, on the genetic variation and population structure of central Argentinian guanacos. Although modern sequencing technologies allow sequencing huge parts of the genome, this study is based on the analyses of only 541 bp D-Loop sequences and 14 microsatellite loci. Nevertheless, the employed methods are sufficient for the purpose of the study and more feasible than analyzing larger genomic regions. The conducted analyses, based on the microsatellite and control region data, are straightforward and sophisticated. The results of this study show high genetic diversity and low levels of inbreeding for guanacos in Argentina, suggesting that the guanaco population is recovering. The population has undergone a bottleneck prior to the arrival of European settlers and colonization further decreased the genetic variation. The results are clearly presented and discussed.
Some specific points:
In line 164 it is stated that the alignment was trimmed to 328 Why? In results 514 bp are mentioned, whereas in Supplementary Table 2 it says 300 bp.
We thank the reviewer for identifying these mistakes. The number of 514bp was corrected in Materials and Methods and in the Complementary Table 2.
Table 1: Several columns need to be widened as the numbers/text in them is split over two lines which is confusing and misleading. Also, the table legend mentions “type of sample” but that is not shown in the actual table.
Done.
Figure 2: The figure legend mentions purple and orange colours in 2C/2D but there is no purple/orange.
Done (lines 325-340).
Lines 518-528: shorten sentences.
Done (line 543-553).
Minor spelling issues:
line 50: more closely (no dash)
Done (line 55).
line 121: well-established
Done (line 129).
line 411: genetic flow
Done (line 423).
line 421: larger populations
Done (line 433).
line 505: suggest that there has been
Done (Line 530).